# Crop Rotation Enhances Agricultural Sustainability: From an Empirical Evaluation of Eco-Economic Benefits in Rice Production

**Dun-Chun He** [1] , **Yan-Li Ma** [1,2], **Zhuan-Zhuan Li** [1,2], **Chang-Sui Zhong** [3], **Zhao-Bang Cheng** [2] **and Jiasui Zhan** [4,*]

[1]  Institute of Eco-technological Economics, School of Economics and Trade, Fujian Jiangxia University, Fuzhou 350108, China; hedc@fjjxu.edu.cn (D.-C.H.); mayl@fafu.edu.cn (Y.-L.M.); lzz@fafu.edu.cn (Z.-Z.L.)
[2]  Institute of Plant Protection, Jiangsu Academy of Agricultural Sciences, Nanjing 210014, China; czb69jaas@jaas.ac.cn
[3]  Agro-Tech Extension and Service Station of Fuqing City, Fuzhou 350300, China; zhengrr@fafu.edu.cn
[4]  Department of Forest Mycology and Plant Pathology, Swedish University of Agricultural Sciences, 75007 Uppsala, Sweden
*   Correspondence: jiasui.zhan@slu.se

**Abstract:** Cropping systems greatly impact the productivity and resilience of agricultural ecosystems. However, we often lack an understanding of the quantitative interactions among social, economic and ecological components in each of the systems, especially with regard to crop rotation. Current production systems cannot guarantee both high profits in the short term and social and ecological benefits in the long term. This study combined statistic and economic models to evaluate the comprehensive effects of cropping systems on rice production using data collected from experimental fields between 2017 and 2018. The results showed that increasing agricultural diversity through rotations, particularly potato–rice rotation (PR), significantly increased the social, economic and ecological benefits of rice production. Yields, profits, profit margins, weighted dimensionless values of soil chemical and physical (SCP) and heavy metal (SHM) traits, benefits and externalities generated by PR and other rotations were generally higher than successive rice cropping. This suggests that agricultural diversity through rotations, particularly PR rotation, is worth implementing due to its overall benefits generated in rice production. However, due to various nutrient residues from preceding crops, fertilizer application should be rationalized to improve the resource and investment efficiency. Furthermore, we internalized the externalities (hidden ecological and social benefits/costs) generated by each of the rotation systems and proposed ways of incenting farmers to adopt crop rotation approaches for sustainable rice production.

**Keywords:** agricultural sustainability; crop rotation; rice; eco-economic benefit; externality

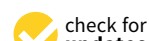

## 1. Introduction

The human population is expected to exceed 9.7 billion by 2050, requiring a substantial increase in agricultural production capacity to secure global food supplies [1] which, on the other hand, are threatened by climate change, environmental pollution and drained natural resources such as water and fossil energy [2,3]. Climate change, environmental pollution and natural resource constraints are also expected to have negative impacts on the productivity and quality of crops. Current agricultural production systems heavily rely on high inputs of natural resources, particularly irrigation water, fertilizers and pesticides. For example, in Samsun, Turkey, the annual energy consumption for wheat production is 35,737 MJ/ha [4]. Up to 15 fungicide sprays are executed annually to control potato late blight in Northern and Western Europe [5] and more than 20 fungicide sprays are applied to control rose mildew in some parts of the world [6]. In apple, more than 12 fungicide applications usually take place each season to control scab caused by *Venturia inaequalis* [7], even though a recent result indicated that only five applications could achieve the similar

control purpose [8]. High resource inputs may usually increase gross production of crops but many of them may not generate positive net returns due to the gain in production being over-weighted by excess inputs. Furthermore, when yield is the primary goal of farmers, which is always the case for cereal productions, little attention will be paid to the direct and indirect effects of the production process on society and ecology such as sustainability of food safety, soil quality and ecological resilience generated by high chemical residues which pollute soils and rivers, demolish biodiversity and poison humans and other animals, etc. Therefore, conservation agriculture as a tool for sustainable development is essential so that natural resources can be used in a rational and economical manner for social and ecological sustainability [9]. Recent concern for the sustainability of agriculture and associated natural environments has led to renewed interest in practices that seek to increase production while improving soil health and ecological resilience [10–13] through crop diversification and comprehensive evaluation of the social, economic and ecological impacts of producing systems [14] by internalizing the externalities, i.e., the hidden benefits and costs are not reflected in marketing prices, associated with primary production systems. Agricultural diversification is referred to the reallocation of some farming resources such as lands, equipment and labor to other social or natural services and can be achieved by multiple paths such as changing cropping systems, modifying productive goals and switching to non-farming activities at spatial and/or temporal scales [15]. Among them, crop rotation has been thought to be a promising agricultural practice which could regenerate balanced biotic and abiotic interactions, supporting a synergistic service to both society and nature by enhancing key elements of biodiversity, increasing resource efficiency, reducing pest epidemics and stabilizing the function of ecosystem production over time [16–18]. For example, it has been shown that crop rotation can eliminate soil-borne pathogens, pests and weed reservoirs that cannot be effectively controlled by pesticides and improve soil quality such as nutrition status and physical structure [19]. However, the benefits associated with crop rotation are rarely evaluated by a comprehensive evaluation of social, economic and ecological impacts of crop rotation generated from field data.

Rice, as one of the main nutrient supplies of the world, is especially important in less developed Asian countries. Soil pollution and ecological deterioration associated with current agricultural production systems greatly threaten sustainable rice production [19]. For example, the projected increase in pore-water arsenite—the more toxic form of arsenic—may cause up to a 39% rice grain reduction compared to current soil arsenic concentrations [20]. Even though higher yields do not always result in better economic benefits [21], the problem of overreliance on chemical inputs while targeting to maximize yield [9] is particularly serious in the rice production system of China. It has been reported that Chinese rice cultivation occupies 20% of global production acreage but consumes 26.7% of chemical nitrogen fertilizers (over 180 kg/ha). Only 20–30% of the fertilizers applied are taken by rice crop [22].

Paddy-upland rotations have received particular attention in the era of agricultural diversification and have been adopted by some regions in Southern China [19] where the available irrigation system allows the farmer's practice to be successfully executed. Although better pay-offs were reported when farmers rotated rice with other crops rather than successively growing rice, how this practice may impact other components of production such as ecology and energy conservation and what the best crop to rotate is are hardly understood. Many governments in the world have focused on increasing the total rice cultivation area through the provision of subsidies to reduce production costs. These economic incentives ensure the steady increase in rice production and encourage farmers to invest more in machinery for rice production, able to handle large acreages. However, their production ambitions and economic returns are not always synchronous, generating concerns of production sustainability. The disagreement results from that fact that net income of rice (as with other crops) production depends not only on immediate, direct factors such as yield, price, subsidy and expenses but also on future, indirect factors such as soil health and ecological resilience. A lack of comprehensive assessment of the synergies

and trade-offs generated by the short-term and long-term interactions between direct and indirect benefits and economic and ecological benefits has resulted in a poor equilibrium among efficiency, cost, profit and sustainability of production. With regards to the research on the rice cropping system, scientists have focused on fundamental questions such as its links with soil chemical and physical properties or applied issues such as technology development rather than social and ecological economics analyses. Particularly, the externalities of rotations for rice production have rarely been quantitatively studied based on data generated from field experiments but are necessary to ensure sustainable rice production to feed the growing global population [23].

In the current study, data generated from fields with different rice cropping systems over two consecutive years were evaluated in parallel with economic, social and ecological effects in order to develop a more profitable, effective and eco-friendly rice production strategy. The specific goals of the study were to (1) determine the differences in the factors responsible for the economic, social and ecological benefits of rice production within different cropping systems; (2) evaluate the pros and cons of rice production among different cropping systems and develop a practice model of rice production in main rice cultivation areas such as Southern China; (3) quantify the externalities of rice production associated with different cropping systems and make recommendations to policy-makers to increase the sustainability of rice production.

## 2. Materials and Methods

### 2.1. Experimental Site

The experimental site (25°33′20.67″ N, 119°25′36.93″ E) was located at the field trial Station of Fujian Agriculture and Forestry University in Jiangjing town, Fuqing city, Fujian province, China. This site has a humid subtropical monsoon climate with mean annual rainfall of 1050~1500 mm and an effective accumulated temperature of 6000–6600 °C, with an average daily temperature of 20–25 °C during growing season. The experimental fields were well equipped with an irrigation system and were either in fallow or planted with watermelon, potato or rice before this study according to the experimental requirements described in the next sections.

### 2.2. Experimental Design and Crop Management

The experiments were conducted between March and August in 2017 and 2018. Each of the field experiments contained treatments including two rice cultivars—Yiyou 673, provided by the Rice Institute, Fujian Academy of Agricultural Sciences, in Fuzhou, and Fulong 3831, provided by the Longyan Institute of Agricultural Science of Longyan City—and four cropping systems (2 × 4), and they were laid out in a completely randomized block design with three replicates (a total of 24 experimental units). The two rice cultivars are similar in many agronomic characteristics such as plant height and maturity and have been widely grown in this region for many years. The four cropping systems were successive rice cropping (RR), fallow followed by rice (FR), potato and rice rotation (PR) and watermelon and rice rotation (WR). Each of the experimental units was 0.2 ha in size and was separated from the others by ~50-cm furrows to prevent water and nutrient flows among units.

The rice seeds were sown in seedling trays in late March. Immediately after sowing, the seedling trays were mulched with white plastic films to maintain temperature and moisture while allowing sunlight to transmit. Experimental fields were prepared by ploughing twice with a power tiller, a harrow and a leveler. An ammonium bicarbonate nitrogen fertilizer (N ≥ 17.1%) (Anhui Liuguo Chemical Co., Ltd., Tongling, China) was applied as a base fertilizer at 450 kg/ha before transplantation according to the theoretical calculation of rice N demand, local average rice yield and the estimated N content in the soil of the experimental fields. Rice seedlings at the stage of 3–4 leaves were transplanted mechanically (Shanghai Kubota Co., Ltd., Shanghai, China) at a density of 165,000–180,000 hills/ha. A compound fertilizer (N:P:K = 16:16:16, total nutrient ≥ 48%, 150 kg/ha) and a urea fertilizer

(N $\geq$ 46.4%, 75 kg/ha) (Anhui Liuguo Chemical Co., Ltd., Tongling, China) were applied at the beginning of the tillering stage. Water, diseases, pests and weeds were managed according to field conditions. The rice was mechanically harvested in August. The rice straws were returned to the fields after grain thrashing.

*2.3. Traits Measurement and Parameters Estimates*

Identical sampling protocols were used for all treatments of the experiments conducted in the two years. Five sample sites were selected from each experimental unit using a stratified strategy with one site in the center of the unit and two sites each in the ends of the unit. Soil samples ($0-15$ cm depth) were collected using a tube auger from the five sampling sites in each experimental unit and were thoroughly mixed to form a composite sample for physical and chemical characterizations [24]. Soil pH, organic matter (SOM), available N, available P and available K were measured by a pH meter, the acidified dichromate method, the alkali hydrolysis and diffusion method, the Olsen method and the atomic absorption spectrophotometry, respectively, using a slurry of 1:2.5 soil/water (*v/v*) as previously described [25–27]. Concentrations of lead (Pb), mercury (Hg), chromium (Cr), cadmium (Cd), copper (Cu) and zinc (Zn) in the soil samples were determined using graphite furnace atomic absorption and flame atomic absorption [28–30]. Straw biomass and grain yield were also determined from the five sampling sites during harvesting and then converted to total production in each of the experimental units using the total areas measured from the five sites (20 m$^2$). Grain production was quantified with all crops in each of the experimental units.

The rice marketing price, governmental subsidy and total production cost associated with farmland rent, consumable materials (e.g., fertilizers, pesticides, seeds, plastic tray and film) and labor (sowing, ploughing, transplanting, fertilizing, managing and harvesting, etc.) were calculated by farm gate price, actual government support and expenses and mechanical devaluation was estimated. To obtain the direct information needed for the calculation, a direct survey involving face-to-face interviews with farmers was conducted as described previously [31]. The survey was conducted with a total of 25 farmers across the five towns of the city. Accordingly, the costs of farmland rent, material and labor in seeding, ploughing, transplanting, fertilization, plant protection (diseases, insects and weeds), harvesting and other miscellaneous expenses were set to 692, 265, 230, 230, 138, 138, 127 and 81 USD per hectare, respectively, with a total cost of 1904 USD per hectare. The annual governmental subsidy for rice cropping was 230 USD per hectare.

Harvest index (HI), revenue (R), profit (NP), profit margin (PM), weighted dimensionless values of soil chemical and physical (SCP) and heavy metal (SHM) traits were calculated using the following formulas [24]:

$$HI = G/(G + DS) \tag{1}$$

$$R = G \times P + S \tag{2}$$

$$NP = R - C \tag{3}$$

$$PM = (NP/C) \tag{4}$$

where G, DS, P, S and C are the grain production, straw weight, grain marketing price, governmental subsidy and total production cost, respectively.

$$SCP = 1/5 \sum_j (x_i - x_{max})/(x_{max} - x_{min}) \tag{5}$$

$$SHM = 1/6 \sum_j (x_i - x_{max})/(x_{max} - x_{min}) \tag{6}$$

where $x_i$, $x_{max}$ and $x_{min}$ are the raw data of each experiment plot and the maximum and minimum raw data of each replication, respectively; $i$ is the experimental plot; $j$ is the order of pH, SOM, N, P and K for SCP and Pb, Hg, Cr, Cd, Cu and Zn for SHM.

The indicators of benefit assessment, including profit, profit margin, revenue, yield, HI, SCP, SHM and weight of dry straw, were determined in line with the documents [32,33] and the expert and farmer consultations as described previously [31]. In total, fifteen experts from Fujian Agriculture and Forestry University, Fujian Academy of Agricultural Sciences, Jiangsu Academy of Agricultural Sciences and the departments of agriculture technology in Fujian and Jiangsu provinces and 25 farmers across the five towns of Fuqing city were consulted for the matters. The indexes (Table S1) of the benefits were weighted using the Analytic Hierarchy Process (AHP) [34] according to their relative importance on the basis of the experiment and the consultation results from the expert and farmer interviews.

To obtain normalization data for the benefits assessment, the raw values of the indicators were converted to dimensionless values $x_i'$ by min-max normalization (Formula (7)) [35]. The benefits index ($BI_i$) of rice production within the different cropping systems was calculated using the Formula (8) [36]:

$$x_i' = (x_i - x_{max})/(x_{max} - x_{min}) \tag{7}$$

where $x_i$, $x_{max}$ and $x_{min}$ are the raw data of indicators from each experiment unit and the maximum and minimum raw data of the corresponding indicators of each replication, respectively; $i$ is the random order of these experimental plots.

$$BI = 1/3\sum\nolimits_j w_i\, x_i' \tag{8}$$

where $w_i$ and $x_i'$ are the weighted and dimensionless values of the $i$th indicator, respectively; $j$ is the random order of the replications. Farmland rent (692 USD/ha) was not included in the economic benefit analysis of the FR practice.

The externality values were calculated as: externality value = profit × (social and ecological benefit index/profit weight × comprehensive benefit index).

### 2.4. Statistical Analysis

The contributions of cropping system, cultivar and their interactions with yield, harvest index, profits and soil properties including pH value and contents of organic matter, minerals and toxin chemicals were assessed using a multivariate analysis of variance (MANOVA), while the contributions of these independent variables to economic, social and ecological benefits as well as externalities were assessed by a one-way ANOVA. In the ANOVA and MANOVA, cultivar was treated as a fixed variable while cropping system was treated as a random variable. Duncan's Multiple Range Test was used to compare means of rice yield, harvest index, soil physical and chemical properties, profits, benefits and externality within dependent variables at the 0.05 probability level. All of the statistical analyses were performed using IBM SPSS 19.0 software (IBM Corp., Armonk, NY, USA).

## 3. Results

### 3.1. Cropping System Significantly Impacts the Socioeconomic Benefits of Rice Production

The ANOVA revealed a significant impact of cropping system on the yield, profit and profit margin of rice production ($p < 0.05$). No cropping system impact on harvest index ($p = 0.335$) was found. Similarly, cultivar and its interaction with cropping system did not have any biological and economic influences on rice production in the current study (Table 1).

**Table 1.** Analysis of variance evaluating the effect of cropping systems, cultivar and their interaction with yield, harvest index and profits of rice production.

| Parameter | Yield | | | Harvest Index | | | Profit | | | Profit Margin | | |
|---|---|---|---|---|---|---|---|---|---|---|---|---|
| | DF | F | P | DF | F | P | DF | F | P | DF | F | P |
| Cultivar | 1 | 0.427 | 0.517 | 1 | 0.603 | 0.442 | 1 | 0.360 | 0.552 | 1 | 0.360 | 0.552 |
| Cropping system | 3 | 3.193 | 0.034 | 3 | 1.166 | 0.335 | 3 | 2.967 | 0.043 | 3 | 2.967 | 0.043 |
| Cultivar × Cropping system | 3 | 0.295 | 0.829 | 3 | 0.511 | 0.677 | 3 | 0.291 | 0.831 | 3 | 0.291 | 0.831 |
| Error | 40 | | | 40 | | | 40 | | | 40 | | |

### 3.2. Difference in Production and Socioeconomic Benefits among Rice Cropping Systems

The yield, profit and profit margin of PR (potato rice rotation), FR (fallow followed by rice) and WR (watermelon rice rotation) were higher than those of RR (successive cropping of rice) (Table 2). Compared with the other three cropping systems, PR achieved the highest yield, profit and profit margin. It was followed by FR while RR performed worst. Rice yield from PR was significantly higher ($p < 0.05$) than that from RR and WR but was only marginally higher than that from FW (Table 2). Profits from PR were also significantly higher than those from all other cropping systems.

**Table 2.** Effect of cropping systems on yield and socioeconomic benefits of rice production.

| Cropping System | Yield t/ha | Harvest Index % | Profit US Dollar/ha | Profit Margin % |
|---|---|---|---|---|
| RR | 5.2 b | 42.2 a | 162 c | 8.5 c |
| FR | 6.1 ab | 45.1 a | 465 b | 24.4 b |
| PR | 7.1 a | 42.8 a | 826 a | 43.4 a |
| WR | 5.9 b | 45.8 a | 385 b | 20.2 b |

Note: The different letters following the values in a column indicate a significant difference ($p < 0.05$). The same letter means it is not significantly different. RR = successive cropping of rice; FR = fallow followed by rice; PR = potato rice rotation; WR = watermelon rice rotation.

### 3.3. Effects of Cropping Systems on the Chemical and Physical Properties of Soils

Chemical and physical properties including pH value, organic matter, mineral and heavy metal contents fluctuated greatly over the sampling times within the growing season (Figures 1 and 2, Table 3) in all cropping systems. Overall, the soils were acidified in the paddy fields and the most acidic soil was found in the RR experiment. Soil organic matter showed a downward trend, especially in WR. N, P and K contents were richest in the soil from PR, leading to the highest SCP index. With the exception of organic matter, WR also yielded better soil fertility (N, P and K) and SCP than those of RR and FR. Regarding the contents of harmful heavy metals, levels under RR were always the highest, although some of the differences were not significant from other cropping systems, leading to the highest SHM (Table 4). The temporal dynamics of the heavy metals in the soils from RR, PR and WR showed a similar trend of slightly increasing over the growing season. This pattern was more obvious in RR (Figure 2, Table 4). Except for Zn, the heavy metal contents in the soils from PR were higher than in those from FR and WR (Table 4).

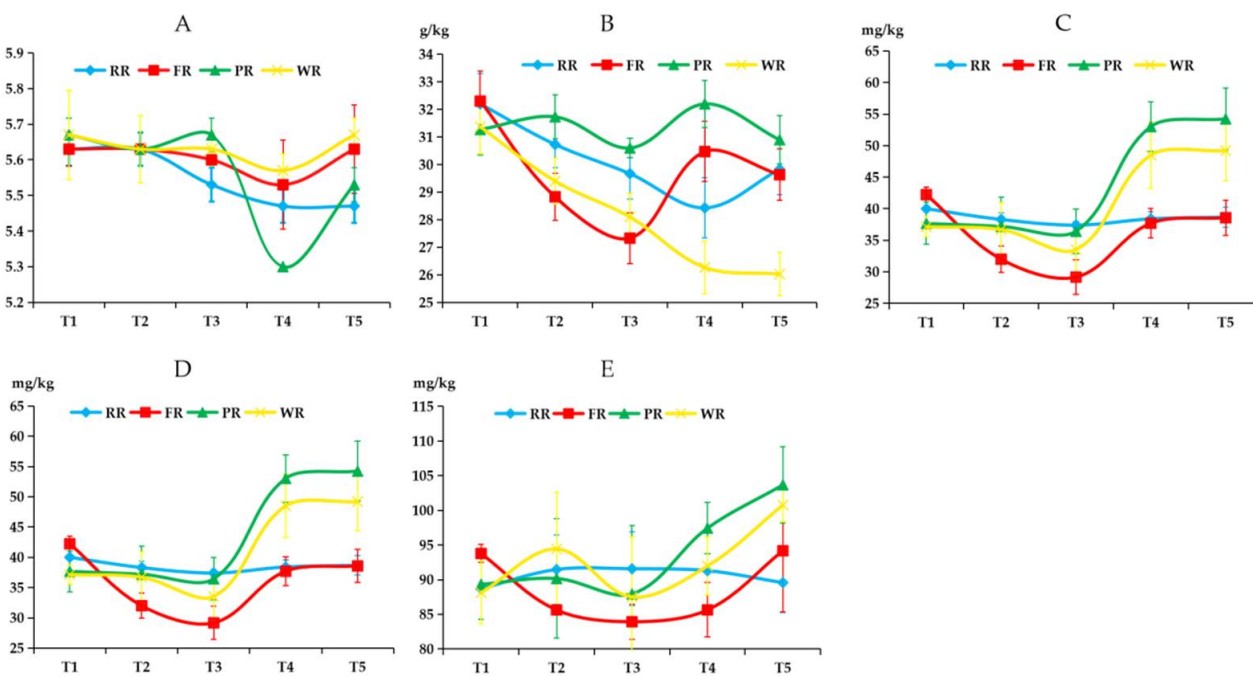

**Figure 1.** The temporal dynamics of soil chemical and physical properties. (**A**) pH value, (**B**) soil organic matter (SOM), (**C**) available nitrogen content (N), (**D**) available phosphorus (P) content and (**E**) available potassium (K) level. RR = successive rice cropping; FR = fallow followed by rice; PR = potato rice rotation; WR = watermelon rice rotation. Sampling date: T1 = September 2016; T2 = March 2017; T3 = September 2017; T4 = March 2018; T5 = September 2018.

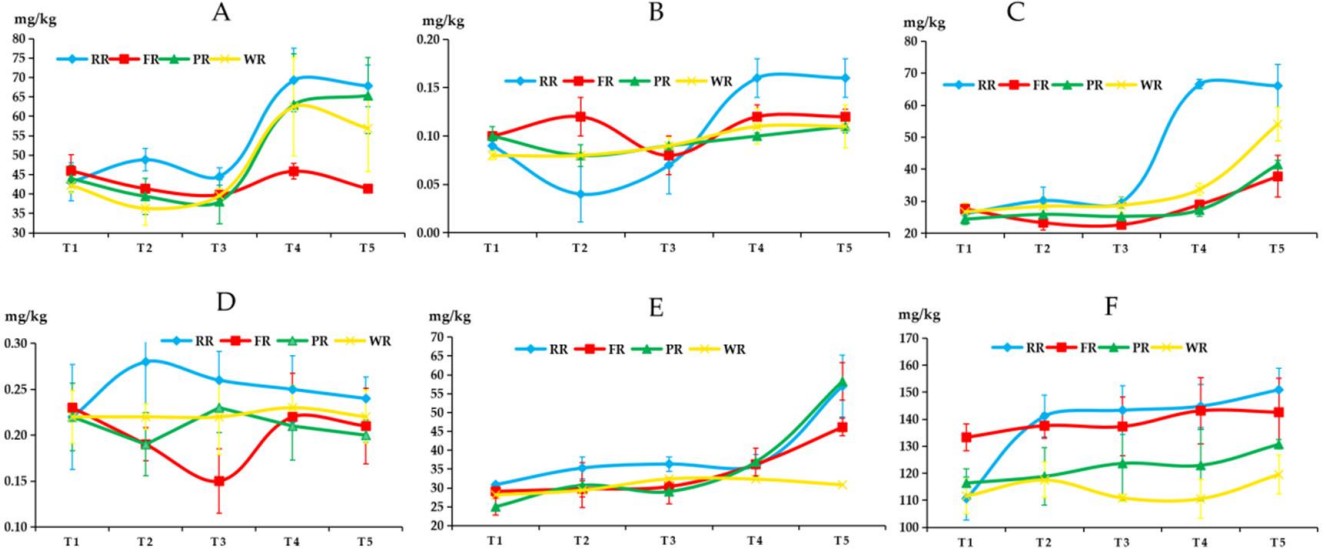

**Figure 2.** The temporal dynamics of soil heavy metal content: (**A**) Pb, (**B**) Hg, (**C**) Cr, (**D**) Cd, (**E**) Cu and (**F**) Zn. RR = successive rice cropping; FR = fallow followed by rice; PR = potato rice rotation; WR = watermelon rice rotation. Sampling date: T1 = September 2016; T2 = March 2017; T3 = September 2017; T4 = March 2018; T5 = September 2018.

**Table 3.** Effect of rice cropping systems on soil pH value, available nitrogen (N), available phosphorus (P), available potassium (K) and organic matter (SOM) level.

| Cropping System | pH | SOM g/kg | N mg/kg | P mg/kg | K mg/kg | SCP |
|---|---|---|---|---|---|---|
| RR | 5.50 b | 29.75 a | 127.00 b | 38.03 ab | 90.57 a | 0.3850 b |
| FR | 5.62 a | 28.48 ab | 131.83 b | 33.87 b | 89.05 a | 0.3833 b |
| PR | 5.60 a | 30.75 a | 156.55 a | 45.33 a | 95.80 a | 0.6850 a |
| WR | 5.65 a | 27.07 b | 132.62 b | 41.37 ab | 94.13 a | 0.5050 ab |

Note: The different letters following the values in a column indicate a significant difference ($p < 0.05$). The same letter means it is not significantly different. RR = successive cropping of rice; FR = fallow followed by rice; PR = potato rice rotation; WR = watermelon rice rotation. SCP is the weighted dimensionless values of pH, SOM, N, P and K. The values presented in the table were calculated from the average of the T1, T2, T3, T4 and T5 values presented in Figure 1.

**Table 4.** Effect of rice cropping systems on the heavy metal contents of soil.

| Cropping System | Pb mg/kg | Hg mg/kg | Cr mg/kg | Cd mg/kg | Cu mg/kg | Zn mg/kg | SHM |
|---|---|---|---|---|---|---|---|
| RR | 56.18 a | 0.11 a | 47.82 a | 0.25 a | 46.70 a | 147.15 a | 0.7650 a |
| FR | 40.60 b | 0.10 a | 30.21 b | 0.18 b | 38.25 ab | 139.97 ab | 0.4000 b |
| PR | 51.70 ab | 0.10 a | 33.40 b | 0.22 ab | 43.67 ab | 127.22 bc | 0.4850 b |
| WR | 48.27 ab | 0.10 a | 38.07 b | 0.22 ab | 31.63 b | 115.23 c | 0.4333 b |

Note: The different letters following the values in a column indicate a significant difference ($p < 0.05$). The same letter means it is not significantly different. RR = successive cropping of rice; FR = fallow followed by rice; PR = potato rice rotation; WR = watermelon rice rotation. SHM is the weighted dimensionless values of Pb, Hg, Cr, Cd, Cu and Zn. The values presented in the table were calculated from the average of the T1, T2, T3, T4 and T5 values presented in Figure 2.

*3.4. Effects of Cropping Systems on Benefits and Externalities of Rice Production*

The economic, social, ecological and comprehensive benefits and externalities generated by PR were always higher, significantly or marginally, than those generated by the other cropping systems, while RR always generated the least benefits (Table 5). FR also generated higher benefits in all aspects, except ecological, than those generated by WR and RR. Relative to RR, we estimated that PR, FR and WR generated 348, 157 and 133 USD/ha externality, respectively (Table 5).

**Table 5.** Effect of rice cropping systems on social, economic and ecological benefits.

| Cropping System | Economic Benefit | Social Benefit | Ecological Benefit | Comprehensive Benefit | Externality Value US Dollar/ha |
|---|---|---|---|---|---|
| RR | 0.1735 b | 0.0537 b | 0.0047 b | 0.2319 b | 0 |
| FR | 0.3028 ab | 0.0872 ab | 0.0083 ab | 0.3984 ab | 157 |
| PR | 0.4009 a | 0.1105 a | 0.0155 a | 0.5269 a | 348 |
| WR | 0.2286 b | 0.0708 b | 0.0100 ab | 0.3094 b | 133 |

Note: The different letters following the values in a column indicate a significant difference ($p < 0.05$). The same letter means it is not significantly different. RR = successive cropping of rice; FR = fallow followed by rice; PR = potato rice rotation; WR = watermelon rice rotation. The benefits were estimated according to the indicators and weights presented in Table S1 and the dimensionless values converted from the raw data using Formula (7). The externality of the RR practice was set to zero (CK) and the externalities of other practices were calculated relative to the RR externality.

## 4. Discussion

*4.1. PR is Worth Implementing on the Basis of Rice Production Benefits*

The cropping system significantly impacts the economic and ecological benefits of rice production (Table 1), and all of the paddy-upland rotations we studied generated better social returns including higher yield, higher profits, higher soil fertility and ameliorated soil contamination than those generated by successive rice cropping (Tables 2–5).

Among them, PR is the best cropping system, supported by the highest economic, social, ecological and comprehensive benefits that it generated (Table 5), consistent with previous reports [37–39]. The farm gate price of potato in the winter cropping areas of Southern China was >0.3 USD/kg over the past 10 years, with an average yield of ~33.5 tons/ha, while the cost of producing potatoes in the same period of time was ~6900 USD/ha, generating a much higher net income in the preceding seasons than growing rice, which was estimated to be 3400 USD/ha [40]. However, yield and economic benefits declined substantially when potatoes were consecutively grown for some years [41]. Taken together, these results indicate that the economic benefit of a PR cropping system outperforms that from an RR or a potato−potato system and could be adopted widely, particularly in Southern China where millions of hectares of arable lands are available in winters after rice crop is harvested [42], and the dry winter there is suboptimal for potato disease epidemics. Ecologically, rice rotation with legumes could be another option in this region, but this practice could not be widely accepted by local farmers due to the small contribution of legumes to the economics of the region. To further enhance the socioeconomic as well as ecological (see below) benefits of rice production, some green manure crops should be intermittently grown after a few cycles of PR practice [43,44].

It was reported that growing watermelon decreased soil fertility [45], but we did not find a general pattern in this regard. WR slightly increased N, P and K levels but marginally or significantly decreased organic matter level in the soils compared to FR and RR (Table 3). In this case, intermittently growing green manure plants after WR practice could, to some extent, compensate the organic matter loss [46,47] WR also generated a mixture of ecological benefits and costs relative to RR.

### 4.2. The Amortized Cost of Fallow Should Be Considered in Production Analysis

FR increased yield, direct farmer income and soil pollution (Tables 2, 4 and 5) but did not impact overall soil fertility (Table 3) compared to the RR system. This falsifies the theoretical expectation of soil fertility restoration associated with the practice. However, fallow can affect the entire soil community structure above and below ground, and its externality cannot be robustly evaluated without a comprehensive study covering a range of topics such as soil fertility, biodiversity, resource consumption, etc. In the current study, we only evaluated the impact of FR on soil nutrient and pollutants using two rice cultivars and further research involving more rice cultivars may be required for a more robust conclusion on the benefits of fallow. Furthermore, fallow practice abandons entire production for one or more seasons and significantly decreases the imminent economic benefit of farmers. This amortized cost should be factored into impact evaluation, resulting in a dilemma between economic and ecological benefits of fallow practice [48]. In spite of some economic and ecological benefits in the production season, the amortized cost of fallow should be considered. Therefore, a substantial government subsidy may be a prerequisite for the practice [49], which may not be sustainable for the countries with limited arable lands and floating cashes to compensate farmers while importing foods in the meantime.

### 4.3. Accurate Management of Water and Fertilizer Could Constitute Supplementary Measures for Rice Production Following Crop Rotation

Rice production heavily relies on high inputs of natural resources such as water and energy required to produce mineral fertilizers [50], greatly threatening the sustainable development of human society. Crop diversification through rotation can improve water as well as nutrient efficiency of rice production as a consequence of increased complementarity in the modes and forms of mineral elements consumed by different crops or crop genotypes [51]. Crop diversification through rotation may also alter soil chemical, physiological and/or biological properties, supporting large and sustainable production [52]. To materialize this advantage, nutritional requirement profiles and preferences of succeeding and preceding crops should be considered jointly. If nutrient residues from the preceding crops are high, the application of fertilizer and other forms of nutrient should be reduced

in succeeding production, and vice versa [51,53]. Together with an appropriate water management strategy, this consideration can reduce ineffective tillers and straw biomass, leading to both improved harvest index and grain yields [54]. The organic matter and mineral element levels in the PR soil were significantly higher than in the other cropping systems (Table 3), suggesting that more nutrient residues are retained in the rotation fields with PR in particular. Therefore, accurate management of water and fertilizer use should constitute important elements of rice production following crop rotation. The highest level of heavy metals in RR (Table 4) also suggests that rotation could ameliorate metal contaminations in paddy soil generated by successive rice cropping and benefit the restoration of soil ecosystems. However, it is not clear whether the heavy metal reduction is due to the enhanced take-up by preceding crops and other biological factors, or using more fertilizers and pesticides in rice or contaminated water for rice irrigation. These issues are worthy of further processing.

*4.4. Externalities and Sunk Costs Are an Important Basis for Making Agricultural Policies*

Farmers usually do not clearly understand the complex quantitative interactions among primary production, input, profit, land use and sustainability [55,56]. The practices they adopt are mainly driven by purely economic factors, particularly the income measured by total production [57]. The risk of production, impacts on following crop and sunk costs associated with short- and long-term externalities such as soil resiliency and ecological sustainability of their lands and surroundings are largely ignored but should be included in decision making. Externalities, regardless of benefits or penalties, will eventually be directed back to producers and societies. As a regulator, governments should use an array of incentives or taxation policies to promote production systems with optimized comprehensive benefits by taking farmer incomes, soil fertility, environment pollution, ecological sustainability and socioeconomic development, etc., into account. In this study, we evaluated the synergistic impact of rice cropping systems on social economics and ecology and found that PR, FR and WR generated 348, 157 and 133 USD/ha externality, respectively. Although we found that rotations helped farmers to generate more profits, they also need to additionally invest in equipment required by different crops. Governments could use some of the externalities generated by rotation to top up the economic benefit of farmers for adopting these cropping systems. In the long term, a subsidy policy can ensure food safety and the protection of ecosystem services [58]. Externalities and sunk costs are an important basis for making agricultural policies; therefore, the inclusion of an externalities subsidy policy was also recommended for ecological production of crops [31]. However, economic policy-makers should evaluate the threshold of the subsidy according to the ecological and social benefits of the practices.

## 5. Conclusions

The overemphasis of farmers on direct output leads to a significant knowledge gap among farmers, governments and researchers [59] and unsustainable socioeconomic systems. This problem could be overcome by creating a dynamic economic policy for the adoption of more reasonable cropping systems by taking into account production externalities [60]. Adopting a cropping system with high positive externalities (ecological and social benefits) would increase natural resource use efficiency and social welfare [61]. Regarding rice production, we showed that yields, profits, benefits and externalities varied significantly among cropping strategies. Paddy-upland rotations, especially PR, showed a clear advantage over successive rice cropping and created substantially positive externalities. Some of the externalities could be directed back to farmers through a subsidy system to compensate their additional investments for equipment. Therefore, externalities and sunk costs should be considered in policy making. The internalization of externalities could be achieved by three ways: (1) cultivation intensification and/or technological advances, such as the precise management of water and fertilizer to increase per unit yield, (2) the appropriate dissemination of information regarding ecological practices and an improve-

ment to the information symmetry of public and private stakeholders, including producers, consumers and material supply services, and (3) the provision of a sufficient subsidy to increase farmers' income to encourage farmers to adopt rational cropping systems.

**Supplementary Materials:** The following are available online at https://www.mdpi.com/2077-047 2/11/2/91/s1, Table S1: The indicators and weights of the benefit assessment for different cropping systems.

**Author Contributions:** Conceptualization, Z.-B.C. and J.Z.; data curation, D.-C.H. and Z.-Z.L.; formal analysis, D.-C.H., Y.-L.M., Z.-Z.L. and J.Z.; funding acquisition, D.-C.H.; investigation, D.-C.H., C.-S.Z. and Y.-L.M.; methodology, D.-C.H. and Y.-L.M.; software, Y.-L.M. and C.-S.Z.; project administration, C.-S.Z.; writing—original draft preparation, D.-C.H.; writing—review and editing, D.-C.H., Z.-B.C., J.Z. and C.-S.Z.; supervision, Z.-B.C. and J.Z. All authors have read and agreed to the published version of the manuscript.

**Funding:** This research was funded by the National Natural Science Foundation of China, grant number 72073028, the Natural Science Foundation of Fujian province, China, grant number 2018J01707, and the Scientific Innovation Foundation of Fujian Agriculture and Forestry University, China, grant number CXZX2018100.

**Institutional Review Board Statement:** Not applicable.

**Informed Consent Statement:** Not applicable.

**Data Availability Statement:** The data presented in this study are available on request from the corresponding author.

**Acknowledgments:** The authors are grateful to the assistance of scientists in Fujian Agriculture and Forestry University, the Institute of Soil and Fertilizer of Fujian Academy of Agricultural Sciences, and the Soil and Fertilizer Station of Fuqing City in Fujian Province, China, in data collection and analysis. We thank Zi-Shuai Wang for his assistance with data collection and experiment implementation. We also thank the experts and farmers for returning the questionnaires and the anonymous reviewers for their valuable comments on the manuscript.

**Conflicts of Interest:** The authors declare no conflict of interest. The funders had no role in the design of the study; in the collection, analyses, or interpretation of data; in the writing of the manuscript, or in the decision to publish the results.

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
