# Peer review of "Crop Rotation Enhances Agricultural Sustainability: From an Empirical Evaluation of Eco-Economic Benefits in Rice Production"

_agriculture, doi:10.3390/agriculture11020091_

Round 1

Reviewer 1 Report

Comments for the authors:

Please find below my suggestions and comments for the manuscript.

Line 16: this is well stated, there is no short term effect regarding crop rotations

Line 21-22: the abbreviations in the brackets are not clear

Line 30: keywords: provide synonyms for the words that exist in the tittle

Line 42-43: I suggest mentioning an apple, which is the king of spraying

Line 46: yield is always the primary goal of farmers, right?

Line 61: here indicate more positive effect of rotations, e.g. effects on pest, microorganisms, weeds, and soil, reducing pesticide application

Line 115: I think it is better to show an average yearly temperature instead of effective accumulated temperature, or provide both values

Line 124: exact value needed for each experimental unit

Paragraph 126-137: there is a lack of important information regarding machinery used, fertilizers, rice varieties (manufacturer, trade name, etc.)

Line 162-163: the authors must present cost in US dollars not in Yuan.

Line 166: the same comment as in the line 21-22

Line 185: check plural of “index”, Table S1? What “S” is?

Paragraph 221-227: give again the explanations of each abbreviation used.  It is way harder to follow because they were mentioned a long ago

Line 228: is enough to present this kind of results using only one decimal point

Line 223: past tense should be applied

Line 237: if not significant therefore no differences. Reword this sentence

Line 241-244: to long sentence, split it

Line 227: the same comment as in the line 237

Line 281: calculate profit in Euros or US Dollars

Besides mentioned comments, results have been presented quite well

The same comments apply to discussion part, units must be converted

Paragraph 283-304: the authors should compare their own results with positive effects of crop rotations found in literature

Line 382-386: is would put this on the begging of the conclusion, then after that other part with yours suggestions for production improvements

Reviewer 2 Report

It is an interesting article that highlight the importance of crop rotation for improving agricultural sustainability. It need to clarify some points before it is accepted for publication.

1) Lines 75-76: Should global land be world rice acreage? It is impossible that
Chinese rice cultivation occupies 20% global land. Please check it.
2) Figure 1 and Figure 2 should include unit too.
3) Lines 337-338: “The excess nutrients may explain the lower harvest index” This explain doesn’t make sense. Authors should know that nutrient is not the only factors that affect grain yield. In the case where is no nutrient stress, high temperature and water stress during flowering and gain filling may limited grain yield and consequently harvest index. Please check the weather to confirm that.
4) Lines 343-344. It is better to check reference to confirm the source of heavy mental at research site. Crop cannot produce heavy mental itself. Paddy fields are likely to receive numerous amounts of anthropogenic pollutants due to overuse of chemical fertilizers and pesticides or heavy mental from irrigated water.

Reviewer 3 Report

The manuscript "Crop rotation enhances agricultural sustainability from an empirical evaluation of eco-economical benefits in rice production" studies the impact of the cropping system on the economic, social, and ecological benefits of rice production. Overall, the manuscript is written well and the results are well presented, followed by meaningful discussion.

1) The author mentioned two different rice cultivars that did not show any significant differences but failed to mention how these two rice cultivars differ. Please explain what are the two types of rice cultivars were used, and why?

2)PR system yielded higher socioeconomic benefits but this is governed by the market price in the production season. How are market price comparable among rice, potato, and watermelon, and is this always the case?

3) Most of the rice is grown in low-land areas that are water-logged or have poor drainage. Are these areas suitable for the PR cropping system or it applies only in upland rice cultivation?

4) Crop rotation is usually recommended with legumes. Are legumes cropping system better than PR, please explain?

5)How are PR system more sustainable (beneficial) if the potato has to be sprayed more frequently against a number of diseases (which is usually the case)? Do you think a further stud is needed to study their environmental impacts?

6) Do you think- just two cropping system is enough to understand the economic, social, and ecological benefits? 

Reviewer 4 Report

This manuscript focused on diversified cropping systems with crop rotation which is economically, ecologically and socially more sustainable than that of continuous rice production. The topic is interesting and relevant. However, I have some suggestions and the authors may consider these to improve the manuscript.

Line 16: Add “,” or “and” after social

Line 120: what’s the name of rice cultivars?

Line 141: “One site in the middle of the unit and two sites in each end of the unit”. Where is other two sites?

These sample sites for what? For soil sample collection or for grain yield measurement?

What is the size of sample sites for grain and straw sample for yield measurements?

Line 150-151:Size of sampling sites?

Line 172:what’s the abbreviation for Q ? No Q in the equation from line 168-171.

Table 2:for 1.9 ton ha-1 (36% more) more yield, why profit is 5 times more in PR compare to RR, when crop management was same for all experimental units ?

Figure 1:In figure 1, two graphs are numbered with C and graph E is not in the figure.

Table 3 & 4:Are the values presented in table 3 & 4 obtained from the mean values of T1, T2, T3, T4 and T5 values present in figure 1&2? Please clarify this. You should present Standard Error (SE) values with mean values.

Line 269-274: This note for which table? Is there any table missing?

Round 2

Reviewer 1 Report

No comments at this point.